# Peer review of "Detection of Label-Free Drugs within Brain Tissue Using Orbitrap Secondary Ion Mass Spectrometry as a Complement to Neuro-Oncological Drug Delivery"

_pharmaceutics, 2022, doi:10.3390/pharmaceutics14030571_

Round 1
Reviewer 1 Report
Dear authors,
I highly appreciate your work and wish you good health and energy to continue. People health (and in particular targeted drug delivery procedures - i.e. for cancer treatment) may benefit from it, once you have established a reliable method to detect, and further quantify certain drugs penetration in various tissues.
Kindly receive below few remarks on the submitted paper, please improve as appropriate to your research approach.
Section 1. Introduction
- I suggest to include few remarks on the drugs and drugs mixtures you chose for this study (reason of choice).
- row 78 "semi-quantitative" seems not supported by presented results (i.e. rows 240-245), detection and discrimination however are clearly demonstrated;
Section 2. Materials and methods
- I suggest to include a sub-section specifically referring to drugs (as you put for Sample prep. and Instrumentation) - and here you may present teh info in rows 94 to 107, together with acronyms used for the drugs (i.e. in Conclusion section - Supplementary material description); also few word on "suitable solvents" mentioned in row 152 (in Results section).
- for the incubation systems used, I suggest to either include temperature stability, or to indicate the specific model employed
- for the OrbiSMIS instrumentation description - a short explanation on what "mode 4" and "mode 7" meaning / capabilities would help the reader to understand the experiment
- Fig.1 - same idea as previous comment - I suggest to either exclude from Figure caption the TOF (time-of-flight) that was not used as one understands, or (better in my opinion) include a short text before the figure where you may explain the choice of Orbitrap vs. TOF.
- Table 1 - similar to previous comment, I agree that for a good understanding of instrument "modes" you present all the available ones; however, a short phrase explaining your choice of "modes" from the available in the instrument will help other researchers to save time and proceed to work benefiting of this previous and study of your team. I suggest to include a text to clarify this aspect.
Section 3. Results
- row 176 - I suggest to introduce a phrase mentioning that Fig. S1 and tables S1-S6 may be found in the Supplementary material
- Fig.2 is hardly legible, both letters and chemical formulas, I suggest splitting
- Fig. 3 - I suggest to mark the drug administration site in the brain tissue images
Kind regards
Reviewer 2 Report
The paper deals with the detection of label-free drugs within brain tissue using OrbiSIMS.
In principle the paper is clear but I have some questions:
- Etoposide, olaparib, gemcitabine, vorinostat and dasatinib are the drugs that were tested in the paper. At one point it is not clear for me what happens with gemcitabine, and vorinostat. The technique may not be as versatile if not applicable to different structures. This part needs to be better described.
- One of the critical points of this work is the fact that the problem of biodistribution and thus the pharmacokinetic aspect is not taken into account. It would be important to have data using the technique to determine the drug concentration after intravenous drug administration. This would also take into account biodistribution and not just the simple penetration of the drug into the brain. The route that a drug takes as it traverses tissue will depend on its physicochemical properties and tumor microenvironment. Many anticancer drugs have limited distribution from blood vessels in solid tumours, which limits their effectiveness.This aspect would highlight more the application and novelty of the technique
- The concentration of the drugs tested is 1mg/ml (1000ppm) which is a very high concentration. Only for olaparib is the detection of 22.4 μg/g comparable to data reported in the literature with other techniques. Which are the concentration of the other drugs?
- The tested drugs present presumably poor water solubility.For some, cyclodextrins are added and I wondered if these might somehow change the absorption of the drugs. How have the other drugs been solubilised?
